# Adaptive Inference for Medical Vision Transformers: Token Reduction or Early Exit?

**Ji Young Byun** [*1]                                                                    JBYUN13@JHU.EDU

**HyunSeo Lee** [*1]                                                                      HLEE267@JHU.EDU

**Jordan Shuff** [1,3,4,5]                                                                 JSHUFF1@JHU.EDU

**Rengaraj Venkatesh** [6]                                                          VENKATESH@ARAVIND.ORG

**Nakul S. Shekhawat**[† 4]                                                             NSHEKHA1@JHMI.EDU

**Kunal S. Parikh**[† 1,3,4,5]                                                                 KSP@JHU.EDU

**Rama Chellappa**[† 1,2]                                                              RCHELLA4@JHU.EDU

[1] *Department of Biomedical Engineering, Johns Hopkins University, Baltimore, MD, USA*

[2] *Department of Electrical and Computer Engineering, Johns Hopkins University, Baltimore, MD, USA*

[3] *Glaucoma Center of Excellence and Center for Nanomedicine, Wilmer Eye Institute, Johns Hopkins University School of Medicine, Baltimore, MD, USA*

[4] *Wilmer Eye Institute, Johns Hopkins University School of Medicine, Baltimore, MD, US*

[5] *Center for Bioengineering Innovation & Design, Johns Hopkins University, Baltimore, MD, USA*

[6] *Aravind Eye Hospital, Pondicherry, India*

**Editors:** Accepted for publication at MIDL 2026

## Abstract

Vision Transformers (ViTs) have demonstrated exceptional performance in medical image analysis, yet their computational demands hinder clinical deployment, particularly in time-sensitive applications. Medical imaging requires sample-adaptive optimization due to dataset heterogeneity across modalities and sample complexity; uniform strategies do not well balance efficiency and accuracy. We propose a unified adaptive inference framework that combines Token Reduction (TR) and Early Exiting (EE) through dataset-specific profiling. Our approach quantifies spatial redundancy via Jensen-Shannon Divergence (JSD) and prediction confidence at intermediate layers to train a lightweight predictor that dynamically selects inference strategies at test time. Across five medical datasets, including a real-world cataract dataset (INSIGHT), our framework achieves 71.4% average floating-point operations (FLOPs) reduction with only 0.1pp accuracy loss, substantially outperforming individual strategies (EE-only: 55.9%, TR-only: 57.7%). On PathMNIST, our adaptive inference framework simultaneously improves accuracy by 1.3pp while reducing computation by 77.2%. On INSIGHT, we maintain baseline accuracy with 69.8% FLOPs reduction, demonstrating robust real-world clinical applicability.

**Keywords:** Vision Transformers, Efficient Inference, Token Reduction, Early Exiting.

---

[*] Contributed equally

[†] Corresponding authors

## 1. Introduction

Vision Transformers (ViTs) have achieved state-of-the-art performance across medical imaging tasks, including dermatological lesion classification (Himel et al., 2024; Al-Waisy et al., 2025), chest X-ray diagnosis (Singh et al., 2024), histopathological tissue analysis (Xu et al., 2023b), and ophthalmic image analysis (Wu et al., 2023), leveraging self-attention to capture long-range dependencies crucial for complex diagnostic tasks (Dosovitskiy et al., 2021). However, clinical deployment faces critical computational barriers in time-sensitive and resource-constrained settings. High-volume screening programs for diabetic retinopathy or cataract screening must process thousands of images daily (Ruamviboonsuk et al., 2022; Tham et al., 2022), where even modest per-image latency accumulates into substantial computational burden. Point-of-care imaging on mobile devices (Xu et al., 2025) operates under severe hardware constraints, making standard ViT models impractical for these scenarios.

Approaches to address computational inefficiency in deep learning divide into model-centric optimizations (quantization (Li and Gu, 2023; Du et al., 2024), compression (Wang et al., 2022b; Zhang et al., 2022), efficient attention (Han et al., 2023)) that apply static architectural changes, and data-centric methods that dynamically adapt to input characteristics. Two prominent data-centric strategies are TR (Rao et al., 2021; Liang et al., 2022; Bolya et al., 2022), which eliminates uninformative tokens, and EE (Bakhtiarnia et al., 2022; Xu et al., 2023a), which terminates inference when samples achieve sufficient confidence. These data-centric approaches are particularly well-suited for medical imaging, where substantial variability exists both across and within modalities (Kline et al., 2022). In safety-critical medical applications, matching the appropriate strategy to dataset-specific characteristics is crucial, as suboptimal choices risk compromised diagnostic accuracy.

To address this gap, we introduce a unified framework for adaptive strategy selection across diverse medical imaging datasets (ISIC2019, PathMNIST, PneumoniaMNIST, RetinaMNIST, INSIGHT). We calibrate dataset-specific thresholds: TR threshold via Jensen-Shannon Divergence between attention distributions, and EE confidence thresholds at checkpoints. At inference, a lightweight CNN predictor estimates redundancy from input images to activate TR while intermediate heads enable early termination based on confidence. This ensures redundant samples undergo TR, high-confidence cases EE, and complex cases receive full processing, maximizing efficiency without compromising diagnostic accuracy.

Our main contributions are:

- **Unified Adaptive Framework:** We propose a unified framework that integrates TR and EE for ViTs, utilizing a lightweight predictor to adaptively activate TR based on input-specific spatial redundancy while leveraging confidence-based EE at intermediate layers, enabling instance-level optimization of both spatial and temporal redundancy.

- **Dataset-Specific Profiling Methodology:** We conduct comprehensive profiling analysis to characterize redundancy-complexity profiles through token-level similarity and sample-wise confidence distribution, revealing that optimal strategies vary across datasets.

- **Superior Efficiency-Accuracy Trade-offs:** Through comprehensive evaluation across five medical imaging datasets, our unified framework achieves 71.4% average FLOPs reduction with 0.1pp average accuracy degradation, outperforming individual strategies (EE-only: 55.9%, TR-only: 57.7%).

## 2. Methodology

This study proposes an integrated framework combining TR and EE to reduce computational cost in ViTs by adapting TR globally based on input image characteristics while using local prediction confidence for EE decisions. Our approach consists of three stages: (1) fine-tuning Data-efficient Image Transformer-Small (DeiT-S) (Touvron et al., 2021) with EE heads on the training set, (2) profiling the validation set to calibrate TR and EE thresholds, and (3) deploying the unified strategy at test-time inference. Detailed dataset statistics are provided in Table A1.

### 2.1. Stage 1: Model Training with Early Exit Heads

We fine-tune DeiT-S with 12 transformer blocks, attaching lightweight classifier heads at layers 4, 7, and 10. Each EE head operates on the CLS token using a two-layer MLP with layer normalization. We train all heads simultaneously using a weighted multi-exit loss:

$$L_{\text{total}} = w_{\text{final}} \cdot L_{\text{final}} + \sum_{k \in \{4,7,10\}} w_k \cdot L_k \tag{1}$$

where $L_k$ is the cross-entropy loss at layer $k$. We set $w_4 = w_7 = w_{10} = 0.3$ and $w_{\text{final}} = 1.0$.

### 2.2. Stage 2: Dataset-Specific Profiling for Strategy Selection

We use the validation set to profile dataset-specific characteristics and calibrate thresholds for adaptive inference: (1) EE confidence thresholds $\theta_{\text{EE}}$ for each checkpoint, (2) ground truth redundancy scores for training the lightweight predictor, and (3) the redundancy threshold $\theta_R$ for TR activation.

#### 2.2.1. EARLY EXIT THRESHOLD CALIBRATION

We calibrate dataset-specific thresholds $\theta_{\text{EE}}$ by sweeping confidence values on the validation set, selecting thresholds that maximize FLOPs reduction while maintaining accuracy degradation <1%. At inference, when confidence $c_k = \max(\text{Softmax}(\mathbf{z}_k))$ at layer $k$ exceeds $\theta_{\text{EE}}$, the sample is classified and inference terminates. Otherwise, computation continues to the next block.

#### 2.2.2. SPATIAL REDUNDANCY PROFILING

To determine which samples benefit from TR, we quantify spatial redundancy using attention similarity. For each validation sample, we compute the ground truth redundancy score $y_{\text{red}}$ based on JSD between attention distributions:

$$y_{\text{red}} = 1 - \frac{1}{3} \left( \text{JSD}(\mathbf{a}_1, \mathbf{a}_4) + \text{JSD}(\mathbf{a}_4, \mathbf{a}_7) + \text{JSD}(\mathbf{a}_7, \mathbf{a}_{10}) \right) \tag{2}$$

where $\mathbf{a}_i$ represents the attention distribution at layer $i$. We select layers to balance coverage and efficiency: early layers alone miss semantic patterns, late layers overlook initial redundancy, while sparser/denser sampling either misses dynamics or adds unnecessary overhead. High $y_{\text{red}}$ values (close to 1) indicate low divergence between attention patterns across layers, suggesting high spatial redundancy where tokens can be safely reduced.

### 2.2.3. Lightweight Redundancy Predictor Training

To avoid the computational overhead of full forward passes at test time, we train a lightweight custom CNN predictor to estimate redundancy from input images. The `Score_Predictor` consists of a feature extractor with three convolutional blocks. The blocks progressively increase channel depth ($32 \rightarrow 64 \rightarrow 128$) using $3 \times 3$ convolutions (stride 2), followed by Batch Normalization and ReLU activation. It predicts $\hat{y}_{\text{red}} \in [0, 1]$ from input image $\mathbf{x} \in \mathbb{R}^{H \times W \times C}$ using mean squared error loss:

$$\mathcal{L}_{\text{pred}} = \frac{1}{N} \sum_{i=1}^{N} (\hat{y}_{\text{red}}^{(i)} - y_{\text{red}}^{(i)})^2 \tag{3}$$

### 2.2.4. Token Reduction Threshold Calibration

We establish the TR activation threshold $\theta_{\text{R}}$ by analyzing the distribution of predicted redundancy scores on the separate validation set. For each validation sample, we first calculate accuracy–FLOPs pairs under identical EE settings for two scenarios: one without TR and one with TR applied at every checkpoint. We then sweep candidate thresholds, using the predicted redundancy score $\hat{y}_{\text{red}}$ to assign each sample to one of the two paths, and sum the previously calculated values to estimate overall accuracy and cost. We select the smallest threshold that maintains accuracy within 1% of the all-token baseline while maximizing FLOPs reduction. At test time, if $\hat{y}_{\text{red}} > \theta_{\text{R}}$, TR is activated after each EE checkpoint (layers 4, 7, and 10).

### 2.3. Stage 3: Unified Inference at Test Time

The complete inference pipeline for each test sample:

1. The `Score_Predictor` estimates $\hat{y}_{\text{red}}$ and sets TR activation flag: `use_tr` $= \hat{y}_{\text{red}} > \theta_{\text{R}}$

2. The ViT processes the image layer-by-layer through the 12 transformer blocks

3. At each checkpoint (layers 4, 7, 10):
   - The EE head computes confidence $c_k = \max(\text{Softmax}(\mathbf{z}_k))$
   - If $c_k > \theta_{\text{EE},k}$, inference terminates and returns $\arg\max(\mathbf{z}_k)$
   - If TR is activated (`use_tr = True`) and $c_k \leq \theta_{\text{EE},k}$, apply TR before continuing

4. If no EE occurs, the final head at layer 12 produces the classification

This unified pipeline enables dataset-adaptive optimization: spatially redundant samples undergo TR, high-confidence samples EE, and challenging samples process through all layers. Algorithm 1 formalizes this complete inference procedure.

## 3. Results and Discussion

### 3.1. Experimental Setup

We evaluate our framework across five public datasets: ISIC2019 (Tschandl et al., 2018; Codella et al., 2018; Hernández-Pérez et al., 2024) (9-class skin lesion classification), PathM-NIST (Yang et al., 2023) (9-class colon tissue classification), PneumoniaMNIST (2-class pneumonia detection), RetinaMNIST (5-class diabetic retinopathy grading), and INSIGHT, a private dataset of anterior segment eye images (4-class cataract classification). All images were resized to a $224 \times 224$ pixels using bicubic interpolation. For training, we applied data augmentation: random affine transformations (rotation range $\pm 10°$, translation up to $10\%$), autocontrast ($p = 0.5$), and horizontal flipping. For testing and validation, images were resized to $224 \times 224$. DeiT-S (Touvron et al., 2021) trained with AdamW optimizer (Loshchilov and Hutter, 2017) (learning rate: $5 \times 10^{-4}$, batch size: 64, epochs: 50).

### 3.2. Dataset Redundancy and Complexity Analysis

We profile dataset-specific redundancy using token-level cosine similarity across DeiT-S layers (Figure 1(a)). RetinaMNIST and ISIC2019 exhibit high initial similarity ($\sim$0.8), while INSIGHT, PathMNIST, and PneumoniaMNIST start at moderate similarity ($\sim$0.6). All datasets show monotonically increasing similarity with depth, confirming progressive feature homogenization (Zhou et al., 2021; Wang et al., 2022a).

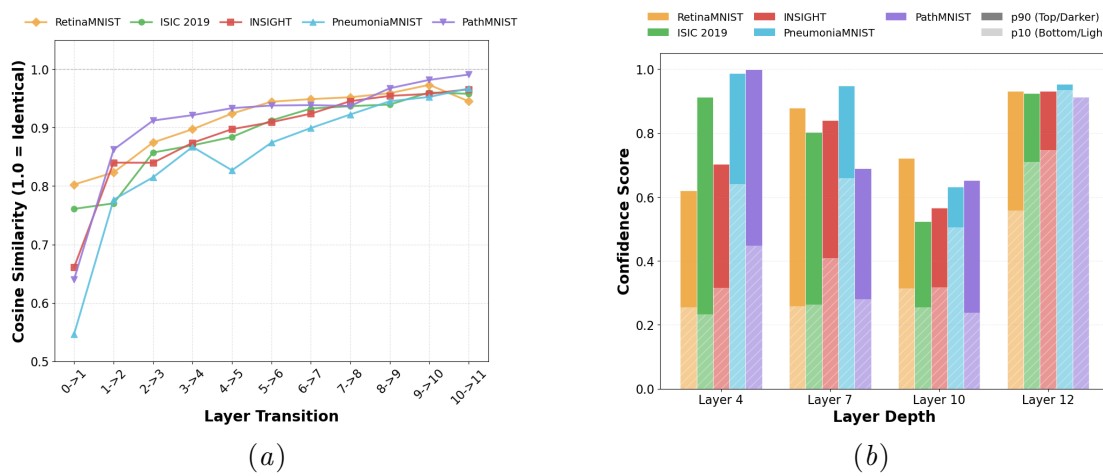

Figure 1: **Dataset redundancy and complexity analysis across DeiT-S layers.** (a) Token-level redundancy across layer transitions. RetinaMNIST and ISIC2019 show high initial similarity ($\sim$0.8); others start lower ($\sim$0.6). All exhibit monotonically increasing similarity. (b) Sample-wise complexity at decision layers (4, 7, 10, 12) showing 90th percentile (easy, lighter) and 10th percentile (hard, darker) confidence. PathMNIST and PneumoniaMNIST achieve high early confidence with minimal easy-hard gaps. RetinaMNIST and INSIGHT show persistent gaps.

Figure 1(b) shows layer-wise confidence evolution at decision layers (4, 7, 10, 12). PathM-NIST and PneumoniaMNIST achieve high early confidence ($>$0.8 by layer 4) with minimal

easy-hard gaps at layer 12, indicating uniform sample complexity. Conversely, RetinaMNIST and INSIGHT start with low confidence and maintain substantial easy-hard gaps through layer 12, reflecting diverse sample complexity. These profiles necessitate adaptive strategy selection: datasets with high early confidence and small gaps (PathMNIST, PneumoniaMNIST) suit aggressive EE, while those with persistent gaps (RetinaMNIST, INSIGHT) benefit more from TR or conservative thresholds.

### 3.3. Dataset-Specific Profiling for Strategy Selection

#### 3.3.1. EARLY EXIT THRESHOLD CALIBRATION

To determine dataset-specific optimal EE thresholds for subsequent experiments, we perform validation set profiling by sweeping $\theta_{\mathrm{EE}} \in [0.5, 0.95]$ to maximize FLOPs reduction while constraining accuracy loss to $< 1\%$. Figure 2 illustrates the performance-efficiency trade-offs for PneumoniaMNIST and INSIGHT (remaining datasets in Figure A1).

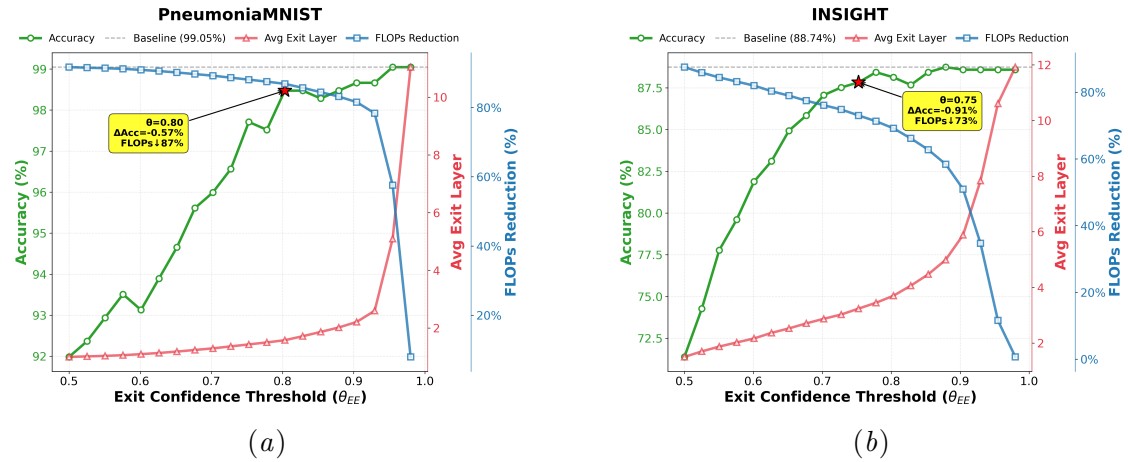

Figure 2: **Performance-efficiency trade-offs of early exiting.** Accuracy (green), FLOPs reduction (blue), and exit layer (pink) vs. confidence threshold. (a) PneumoniaMNIST achieves 87% FLOPs reduction at $\theta_{\mathrm{EE}} = 0.80$ (average exit layer 1.58, $-0.57$pp from baseline). (b) INSIGHT requires $\theta_{\mathrm{EE}} = 0.75$ (average exit layer 3.24, $-0.61$pp from baseline) for 73% FLOPs reduction.

PneumoniaMNIST (Figure 2(a)) achieves 87% FLOPs reduction at $\theta_{\mathrm{EE}} = 0.80$ (average exit layer 5.8, $-0.57$pp from baseline). INSIGHT (Figure 2(b)) requires $\theta_{\mathrm{EE}} = 0.75$, exiting at layer 10.6 for 73% reduction ($-0.61$pp from baseline). This disparity confirms that datasets with high early-layer similarity enable aggressive EE, while those with diverse initial representations require deeper inference. Based on this profiling, we select: $\theta_{\mathrm{EE}}^{\mathrm{Retina}} = 0.78$, $\theta_{\mathrm{EE}}^{\mathrm{Pneumonia}} = 0.80$, $\theta_{\mathrm{EE}}^{\mathrm{INSIGHT}} = 0.75$, $\theta_{\mathrm{EE}}^{\mathrm{ISIC}} = 0.65$, $\theta_{\mathrm{EE}}^{\mathrm{Pathology}} = 0.60$.

#### 3.3.2. TOKEN REDUCTION KEEP RATE SELECTION

Figure 3 presents the accuracy-efficiency trade-offs of various TR strategies—random pruning, Top-K pruning, EViT (Liang et al., 2022), A-ViT (Yin et al., 2022), and Token Merging

(ToMe) (Bolya et al., 2022)—as a function of average token count across all DeiT-S layers, revealing substantial differences in robustness across datasets.

PathMNIST demonstrates resilience (Figure 3(a)): all strategies maintain performance above 99.6%, even with aggressive reduction to 16 tokens (ToMe) versus the 99.91% baseline. Random, Top-K, and EViT maintain ∼99.9% accuracy across all token budgets. INSIGHT shows distinct sensitivity (Figure 3(b)): while Random pruning and EViT preserve ∼87% accuracy down to 76 tokens, Top-K drops sharply to 84.0% at 100 tokens (3.2% loss from the 87.2% baseline). This validates Figure 1(a): PathMNIST's higher token similarity indicates more redundant spatial information, which is precisely what TR exploits: when tokens are similar, fewer are needed to preserve discriminative features.

Validation set profiling identifies EViT as the most stable strategy with minimal sensitivity to token budgets (Figure A2). The optimal keep rates (the proportion of tokens preserved at each layer) vary: ISIC2019/RetinaMNIST: 0.3, PneumoniaMNIST: 0.4, INSIGHT: 0.5, PathMNIST: 0.7. For fair comparison across TR-only and TR+EE configurations, we standardize at keep rate 0.4 for all experiments. At 40 retained tokens, EViT delivers a consistent ∼56% FLOPs reduction (2.027 vs. 4.608 GFLOPs baseline) with competitive accuracy: PathMNIST 99.89% (−1.2pp from baseline), PneumoniaMNIST 97.9% (+2.0pp), INSIGHT 86.3% (−0.9pp), RetinaMNIST 60.0% (+1.0pp), ISIC2019 56.78% (+2.6pp).

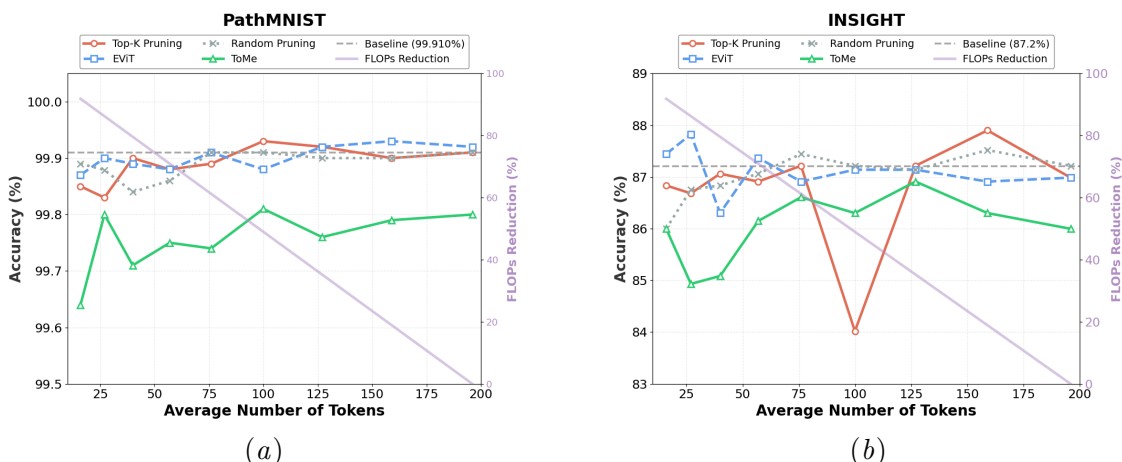

Figure 3: **Token reduction strategy comparison across medical imaging datasets.** X-axis: average token count; y-axis: accuracy (%, left) and FLOPs reduction (%, right, purple). Methods: Top-K (red circles), EViT (blue squares), Random (gray crosses), ToMe (green triangles). Gray dashed line: baseline accuracy. (a) PathMNIST: All strategies maintain > 99.7% accuracy at 40 tokens; EViT achieves 99.89% (−1.2pp). (b) INSIGHT: EViT shows superior stability, maintaining 86.3% at 40 tokens (−0.9pp), while Top-K and Random exhibit higher variance. ToMe collapses below 76 tokens.

### 3.4. Lightweight Redundancy Predictor Performance

Table 1 demonstrates that the lightweight `Score_Predictor` effectively approximates spatial redundancy. RetinaMNIST, PathMNIST, and INSIGHT achieve strong performance (MAE $\leq 0.082$, Pearson $R \geq 0.70$), while ISIC2019 and PneumoniaMNIST show weaker correlations (Pearson $R = 0.48$ and $0.31$) due to narrower dynamic ranges in their redundancy distributions. Nevertheless, the predictor remains sufficiently accurate for coarse separation between low- and high-redundancy samples to trigger TR decisions. The calibrated thresholds reveal dataset-specific regimes: RetinaMNIST exhibits the highest threshold ($\theta_R = 0.9431$) to preserve subtle vascular patterns, while PathMNIST and INSIGHT adopt permissive thresholds ($\theta_R = 0.0870$ and $0.1883$) consistent with higher baseline redundancy, enabling dataset-aware efficiency without expensive redundancy estimation at test time.

Table 1: **Lightweight redundancy predictor performance on validation data.** The `Score_Predictor` estimates spatial redundancy scores to determine TR activation. Mean absolute error (MAE), Pearson R, and $R^2$ evaluate prediction performance. Optimal thresholds $\theta_R$ are calibrated per dataset to balance computational savings and accuracy.

| Dataset | MAE ($\downarrow$) | Pearson R ($\uparrow$) | $R^2$ ($\uparrow$) | Optimal Threshold ($\theta_R$) |
|---|---|---|---|---|
| ISIC 2019 | 0.1352 | 0.4756 | 0.2124 | 0.1952 |
| PneumoniaMNIST | 0.1631 | 0.3055 | 0.0730 | 0.2988 |
| RetinaMNIST | 0.0804 | 0.7917 | 0.3567 | 0.9431 |
| PathMNIST | 0.0635 | 0.7039 | 0.4701 | 0.0870 |
| INSIGHT | 0.0818 | 0.7408 | 0.4932 | 0.1883 |

### 3.5. Unified Framework Performance

Table 2 compares our unified TR+EE framework against baseline and individual strategies across five datasets. By combining spatial and depth-wise pruning, our approach maintains near-baseline accuracy while processing an average of only 46.0 tokens with an average exit layer of 6.5.

The results highlight distinct behaviors across datasets, validating the need for dataset-specific profiling. PathMNIST achieves 96.0% accuracy (+1.3pp), as TR removes redundant background tokens enabling EE to focus on salient tissue structures and exit early (avg. layer 3.0). Similarly, the real-world INSIGHT dataset maintains baseline accuracy (86.2%), exploiting the high spatial redundancy typical of anterior segment imaging (avg. 29.2 tokens).

However, ISIC2019 and PneumoniaMNIST show minor degradation. ISIC2019's 2.5pp loss stems from aggressive TR at layer 10.3, eliminating subtle fine-grained texture that the model may rely on for diagnosis. PneumoniaMNIST's 1.3pp loss indicates premature termination on subtle cases. While TR-only improved accuracy (92.1%), the unified framework's aggressive EE prevents token-reduced representations from reaching deeper layers where they could recover performance. Both cases demonstrate the inherent trade-off: achieving extreme architectural sparsity requires a careful balance between the speed of termination and the preservation of fine-grained pathological features.

Table 2: **Unified Framework Performance Across Datasets.** All methods use DeiT-S backbone. Baseline: 196 tokens, 12 layers. EE-only: 196 tokens with dynamic EE (dataset-specific $\theta_{EE}$). TR-only: EViT with 40 tokens across all layers. A-ViT: state-of-the-art adaptive ViT with 196 tokens. TR+EE (Ours, shaded): combines TR and EE. Best results in **bold**.

| Dataset | Strategy | Accuracy (%) | Avg Tokens | Avg Exit Layer |
|---|---|---|---|---|
| **ISIC2019** | Baseline | 54.2 | 196 | 12.0 |
| | EE-only | **56.8** (↑2.6pp) | 196 | **9.26** |
| | TR-only | **56.8** (↑2.6pp) | 40 | 12.0 |
| | A-ViT | **56.8** (↑2.6pp) | 196 | 12 |
| | TR+EE | 51.7 (↓2.5pp) | **20.8** | 10.3 |
| **PneumoniaMNIST** | Baseline | 90.1 | 196 | 12.0 |
| | EE-only | 87.8 (↓2.3pp) | 196 | **1.86** |
| | TR-only | **92.1** (↑2.0pp) | **40** | 12.0 |
| | A-ViT | 92.0 (↑1.9pp) | 196 | 12 |
| | TR+EE | 88.8 (↓1.3pp) | 56.3 | 4.65 |
| **RetinaMNIST** | Baseline | 59.0 | 196 | 12.0 |
| | EE-only | **61.0** (↑2.0pp) | 196 | **5.45** |
| | TR-only | 54.5 (↓4.5pp) | **40** | 12.0 |
| | A-ViT | 57.0 (↓2.0pp) | 196 | 12.0 |
| | TR+EE | 60.8 (↑1.8pp) | 44.8 | 7.7 |
| **PathMNIST** | Baseline | 94.7 | 196 | 12.0 |
| | EE-only | 94.4 (↓0.3pp) | 196 | 11 |
| | TR-only | 93.5 (↓1.2pp) | **40** | 12.0 |
| | A-ViT | 92.0 (↓2.7pp) | 196 | 12.0 |
| | TR+EE | **96.0** (↑1.3pp) | 79 | **3.0** |
| **INSIGHT** | Baseline | 86.1 | 196 | 12.0 |
| | EE-only | 87.4 (↑1.3pp) | 196 | 7.04 |
| | TR-only | 85.5 (↓0.6pp) | 40 | 12.0 |
| | A-ViT | 84.9 (↓1.2pp) | 196 | 12.0 |
| | TR+EE | **86.2** (↑0.1pp) | **29.2** | **6.9** |
| *Average Performance Across All Datasets* | | | | |
| | EE-only | −0.3pp | 196 | 7.0 |
| | TR-only | −0.4pp | **40** | 12.0 |
| | A-ViT | −0.28pp | 196 | 12.0 |
| | **TR+EE** | **-0.1pp** | 46.0 | **6.5** |

Per-class analysis (Table A2) reveals that our framework can improve sensitivity beyond baseline for diagnostically challenging classes, such as PathMNIST's mucus (98.16%) and cancer-associated stroma (72.92%), demonstrating that adaptive inference enhances performance on complex tissue types.

We conducted an ablation study on backbone architecture by evaluating ViT-S (Table A3). TR+EE achieves 54.1% to 76.4% FLOPs reduction across five datasets with accuracy changes ranging from -4.5pp to +0.2pp relative to baseline. These results suggest that

architectural characteristics influence the efficiency-accuracy trade-off, informing model selection for clinical deployment with varying computational and accuracy requirements.

### 3.6. Computation Cost Analysis

Table 3 evaluates the computational cost of our unified framework across five datasets. Our approach achieves a significant reduction in algorithmic complexity, averaging a 71% decrease in GFLOPs across all datasets. This substantially outperforms A-ViT, which only achieves an average reduction of 28.9%. These results highlight the advantages of combining token reduction to lower the cost per layer with early exiting to reduce the total depth of the network.

Table 3: **Computation Cost Analysis Across Datasets.** Comparison of theoretical complexity (FLOPs), real-world latency, and energy consumption. All methods use DeiT-S backbone. Speedup is calculated relative to the Baseline of each dataset. Best results in **bold**.

| Dataset | Strategy | FLOPs (G) | Latency (ms) | Energy (mJ) | Speedup |
|---|---|---|---|---|---|
| **ISIC2019** | Baseline | 4.61 | 1.201 | 9.776 | 1.00× |
| | EE-only | 2.616 (↓43.3%) | 1.252 (↑4.2%) | 10.552 (↑8.0%) | 0.96× |
| | TR-only | 2.027 (↓56.0%) | 0.552 (↓54.0%) | 4.654 (↓52.3%) | 2.18× |
| | A-ViT | 3.742 (↓18.8%) | 1.431 (↑19.2%) | 11.724 (↑20.0%) | 0.84× |
| | TR+EE | **1.569** (↓**66.0%**) | **0.629** (↓47.6%) | **4.886** (↓50.0%) | **1.91×** |
| **Pneumonia** | Baseline | 4.61 | 1.188 | 9.404 | 1.00× |
| | EE-only | **0.776** (↓**83.2%**) | **0.396** (↓**66.7%**) | **3.355** | **3.00×** |
| | TR-only | 2.027 (↓56.0%) | 0.543 (↓54.3%) | 4.324 (↓54.0%) | 2.19× |
| | A-ViT | 3.032 (↓34.2%) | 1.414 (↓19.0%) | 11.250 (↑19.6%) | 0.84× |
| | TR+EE | 1.204 (↓73.9%) | 0.575 (↓51.6%) | 4.503 (↓52.1%) | 2.07× |
| **RetinaMNIST** | Baseline | 4.61 | 1.189 | 9.116 | 1.00× |
| | EE-only | 1.662 (↓63.9%) | 0.812 (↓31.7%) | 199.592 (↑2089.4%) | 1.46× |
| | TR-only | 2.027 (↓56.0%) | 0.546 (↓54.1%) | 4.415 (↓51.6%) | 2.18× |
| | A-ViT | 4.320 (↓6.3%) | 1.420 (↑19.4%) | 353.233 (↑3774.9%) | 0.84× |
| | TR+EE | **1.398** (↓**70.0%**) | **0.620** (↓47.9%) | **4.853** (↓46.8%) | **1.92×** |
| **PathMNIST** | Baseline | 4.61 | 1.225 | 12.401 | 1.00× |
| | EE-only | 3.049 (↓33.9%) | 1.335 (↑9.0%) | 2.098 (↓83.1%) | 0.92× |
| | TR-only | 2.027 (↓56.0%) | 0.575 (↓53.1%) | 4.979 (↓59.9%) | 2.13× |
| | A-ViT | 2.386 (↓48.2%) | 1.468 (↑19.8%) | 12.611 (↓1.7%) | 0.83× |
| | TR+EE | **1.050** (↓**77.2%**) | **0.475** (↓61.2%) | **4.243** (↓65.8%) | **2.58×** |
| **INSIGHT** | Baseline | 4.61 | 1.196 | 9.776 | 1.00× |
| | EE-only | 2.061 (↓55.3%) | 2.129 (↑78.0%) | 9.796 (↑0.2%) | 0.56× |
| | TR-only | 1.633 (↓64.6%) | **0.548** (↓54.1%) | 4.486 (↓54.1%) | **2.18×** |
| | A-ViT | 2.902 (↓37.0%) | 1.421 (↑18.7%) | 12.561 (↓28.5%) | 0.84× |
| | TR+EE | **1.394** (↓**69.8%**) | 0.631 (↓47.2%) | 4.568 (↓53.3%) | 1.90× |

Although edge device deployment involves distinct engineering such as quantization, we provide GPU latency and energy measurements to demonstrate tangible efficiency gains

and to validate improvements in computational efficiency. Our framework achieves a $2.07\times$ average speedup over the baseline and a decrease in latency of 51.2%. We also achieve an average of 54.3% energy reduction while maintaining highly stable consumption of 4.6 mJ across all modalities. This stability is particularly evident in RetinaMNIST, where both EE-only and A-ViT exhibit a sharp increase in energy, reaching 199.6 mJ and 353.2 mJ respectively.

### 3.7. Visualization and Failure Case Analysis

Figure 4 visualizes the adaptive framework behavior on representative samples from IN-SIGHT and PathMNIST datasets. Figure $4(a)$ demonstrates sequential TR across all checkpoints with 40% retention rate (keep rate = 0.4), where the model processes all layers for prediction while successfully preserving informative regions around the pupil even at the final checkpoint. Figure $4(b)$ shows a case where TR activates at the first checkpoint followed by EE, as the clear evidence of mature cataract at the pupil enables confident prediction without deeper processing. Similarly, Figure $4(c)$ illustrates a PathMNIST sample where TR activation at the first checkpoint is followed by EE, demonstrating the framework's ability to adaptively combine both efficiency strategies based on sample characteristics.

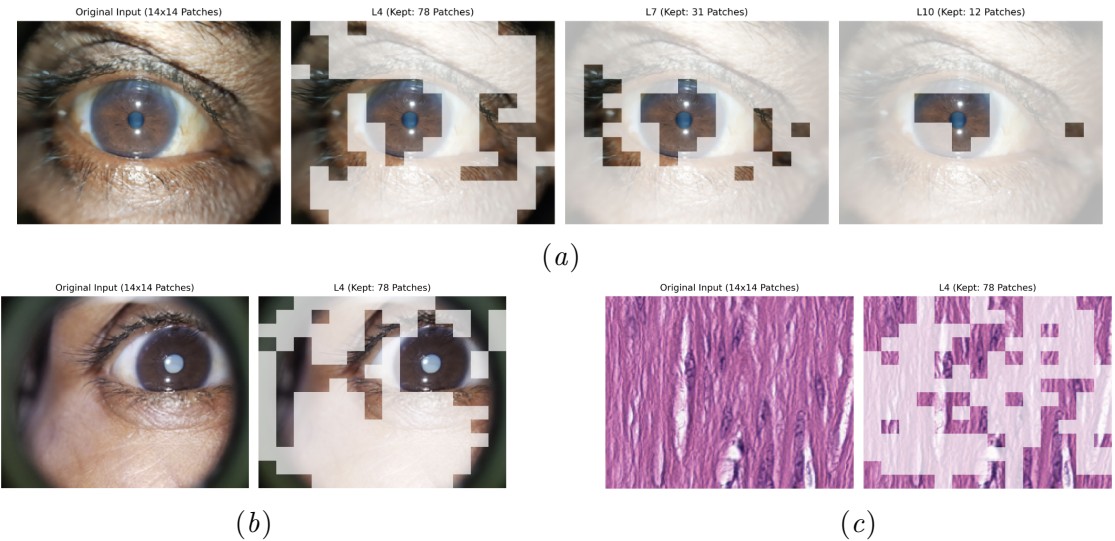

Figure 4: **Visualization of adaptive framework on medical imaging samples.** (a) TR-only with sequential token reduction maintaining diagnostic regions across all layers. (b), (c) Combined TR+EE examples from INSIGHT and PathMNIST.

Figure 5 illustrates the primary failure modes encountered by our unified framework, specifically within the ISIC2019 and PneumoniaMNIST datasets. In ISIC2019, the performance degradation is primarily driven by extreme class imbalance, where the dominant nevus class (50.8%) biases threshold calibration toward majority features. As shown in Figure $5(a)$, aggressive token reduction prunes subtle diagnostic textures critical for rare classes like dermatofibroma (0.9% of the dataset), leading to a 10.78pp specificity gain in majority classes but a sensitivity drop in rare classes. Conversely, PneumoniaMNIST failures are largely

attributed to model overconfidence. Figures 5(b) and 5(c) show how the early exit mechanism triggers on erroneous high-confidence predictions at layers 4 or 7. This overconfidence is compounded by the TR and EE interaction, where token-reduced representations (averaging 56.3 tokens) require the deeper architectural processing of all 12 layers to maintain accuracy, yet the EE mechanism terminates inference at an average layer of 4.65.

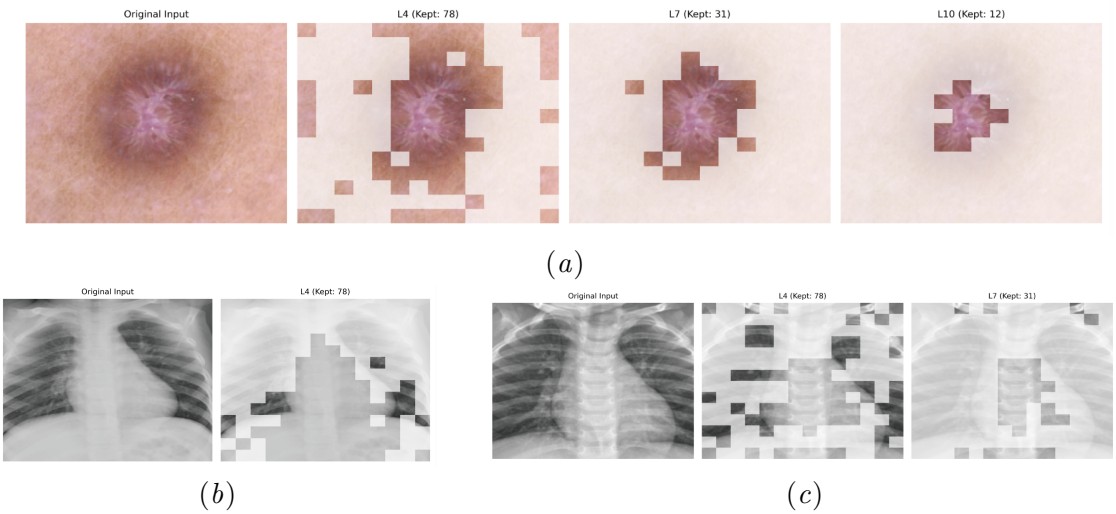

Figure 5: **Qualitative failure-case analysis of adaptive framework** (a) ISIC Dermatofibroma sample with misclassification. (b), (c) Overconfidence failures in PneumoniaMNIST where normal scans were misidentified due to premature termination at layers 4 and 7, respectively.

## 4. Conclusion

This work introduces a unified framework that integrates TR and EE for ViT inference in medical imaging. We calibrate dataset-specific thresholds: prediction confidence for EE ($\theta_{EE}$) and spatial redundancy via JSD for TR ($\theta_R$). At test time, a lightweight CNN predictor estimates sample-level redundancy to activate TR, while intermediate classifier heads enable EE based on confidence. Across five diverse datasets, our framework achieves 71.4% average FLOPs reduction while maintaining diagnostic accuracy within 0.1pp of baseline, substantially outperforming individual strategies (EE-only: 55.9%; TR-only: 57.7%).

*Clinical Impact:* Medical AI deployment demands both accuracy and efficiency, particularly in resource-constrained and time-sensitive settings. Our framework achieves substantial efficiency gains without compromising performance on both public benchmarks and real-world clinical data with inherent quality variability. This approach can enable broader access to diagnostic AI where hardware resources are scarce but patient need is greatest.

*Limitations:* Our framework requires dataset-specific profiling to calibrate thresholds, but this one-time overhead maximizes test-time efficiency without recurring costs. Beyond theoretical FLOPs reductions, validation on edge devices is necessary to assess actual latency improvements.

## Acknowledgments

We acknowledge support from the National Eye Institute (P30EY001765, R21EY034343), VentureWell Propel Award, Microsoft Acceleration Award, Stephen F Raab and Mariellen Brickley-Raab Rising Professorship in Ophthalmology, Johns Hopkins University, and the National Academy of Medicine. In addition, funds to support this AITC study were provided by the Johns Hopkins University AITC under award number P30AG073104. Ji Young Byun was supported in part by a discretionary fund at Johns Hopkins University's Whiting School of Engineering.

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

Table A1: Dataset statistics and data split information. The validation set is split into two equal parts: 50% for dataset-specific profiling and lightweight predictor training, and 50% for token reduction threshold calibration.

| Dataset | Train | Validation | Test |
|---|---|---|---|
| ISIC2019 | 1791 | 448 | 118 |
| PneumoniaMNIST | 4,708 | 524 | 624 |
| RetinaMNIST | 1,080 | 120 | 400 |
| PathMNIST | 89,996 | 10,004 | 7,180 |
| INSIGHT | 5256 | 657 | 657 |

*INSIGHT Ethics Statement* After obtaining informed consent, community health workers collect smartphone-based images of patients attending community eye screenings. Diagnosis labels for each image were obtained using clinical diagnoses made via pen light examination by ophthalmologists at the same screening. The study was approved by the Institutional Review Boards of Aravind Eye Hospital and the Johns Hopkins University School of Medicine.

---

**Algorithm 1** Unified Adaptive Inference Pipeline (TR + EE)

---

**Input:** Input Image $\mathbf{x}$, Redundancy Threshold $\theta_{\mathrm{R}}$, EE Thresholds $\{\theta_{\mathrm{EE}}\}$,
       Token Keep Rate $r$

**Output:** Prediction $\hat{y}$, Exit Layer $k^*$

```
// Stage 1:  Adaptive Token Reduction (TR) Activation
```
$\hat{y}_{\mathrm{red}} \leftarrow \mathtt{Score\_Predictor}(\mathbf{x})$                           `// Predict redundancy score`
$\mathtt{use\_tr} \leftarrow (\hat{y}_{\mathrm{red}} > \theta_{\mathrm{R}})$                           `// Activate TR if redundant`
$\mathbf{Z} \leftarrow$ Initial Tokens with $N_0$ tokens

```
// Stage 2:  Iterative Layer Processing with TR and EE
```
**for** $k \leftarrow 0$ **to** $11$ **do**
    $\mathbf{Z} \leftarrow \mathrm{Transformer\_Block}_k(\mathbf{Z})$
    **if** $k \in \{3, 6, 9\}$ **then**
        `// Early Exit Check`
        $\mathbf{z}_k \leftarrow \mathrm{CLS\_Token\_Output}(\mathbf{Z})$   $c_k \leftarrow \max(\mathrm{Softmax}(\mathbf{z}_k))$
        **if** $c_k > \theta_{EE}$ **then**
            **return** $\hat{y} \leftarrow \arg\max(\mathbf{z}_k)$, $k^* \leftarrow k$
        **end**
        `// Token Reduction (if active)`
        **if** $use\_tr = True$ **then**
            $N_{\mathrm{new}} \leftarrow N_{\mathrm{current}} \cdot r$   $\mathbf{Z} \leftarrow \mathrm{Reduce\_Tokens}(\mathbf{Z}, N_{\mathrm{new}})$
        **end**
    **end**
**end**

```
// Stage 3:  Default (Full Depth)
```
$\mathbf{z}_{12} \leftarrow \mathrm{CLS\_Token\_Output}(\mathbf{Z})$   **return** $\hat{y} \leftarrow \arg\max(\mathbf{z}_{12})$, $k^* \leftarrow 12$

---

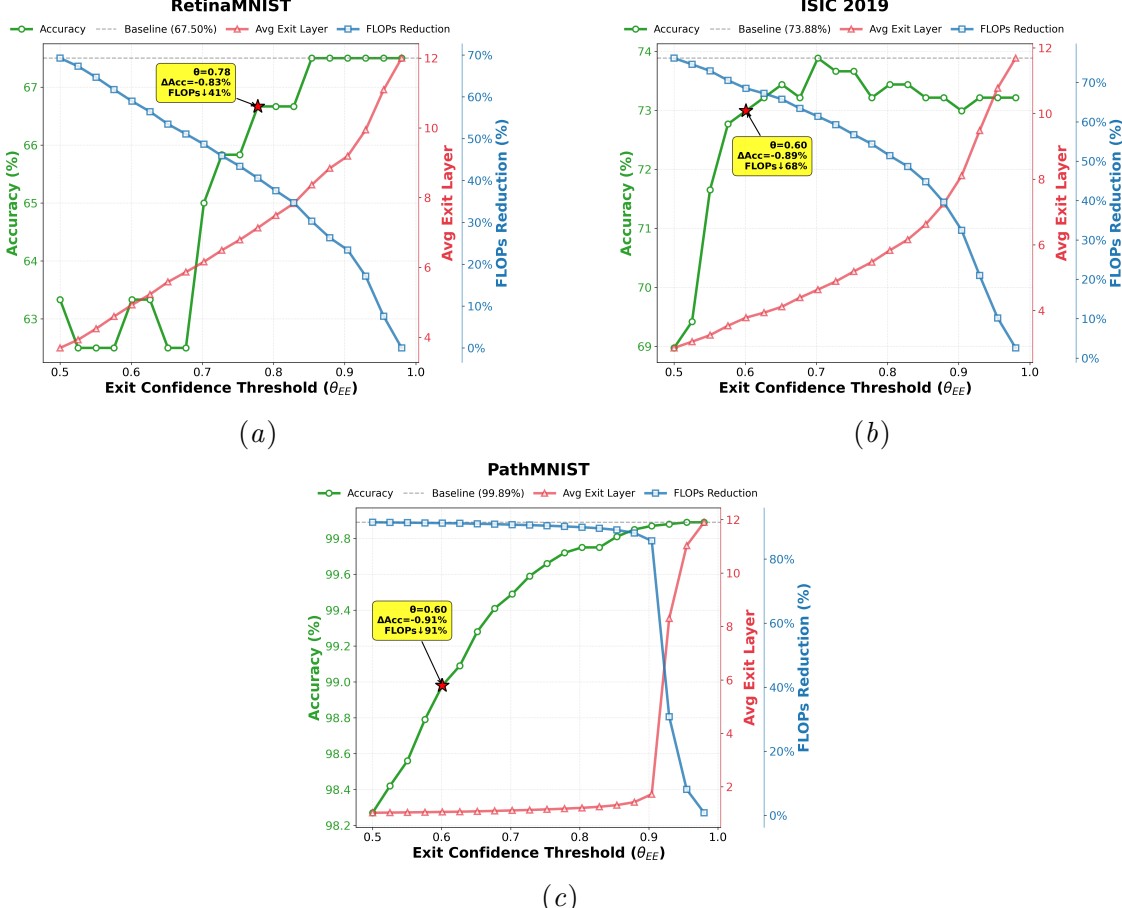

Figure A1: Performance-efficiency trade-offs of early exiting across confidence thresholds. Validation set profiling sweeps $\theta_{\text{EE}} \in [0.5, 0.95]$ to identify optimal thresholds maximizing FLOPs reduction while constraining accuracy loss to $< 1\%$. X-axis shows confidence threshold $\theta_{\text{EE}}$; left y-axis shows accuracy (%, green circles), right y-axis shows FLOPs reduction (%, blue squares) and average exit layer (pink triangles). (a) RetinaMNIST: Achieves 41% FLOPs reduction at $\theta_{\text{EE}} = 0.78$ with 66.67% accuracy ($-0.83$pp from baseline). (b) ISIC2019: Achieves 65% FLOPs reduction at $\theta_{\text{EE}} = 0.65$ with 72.32% accuracy ($-0.89$pp from baseline). (c) PathMNIST: Achieves 91% FLOPs reduction at $\theta_{\text{EE}} = 0.60$ with 98.98% accuracy ($-0.91$pp from baseline)

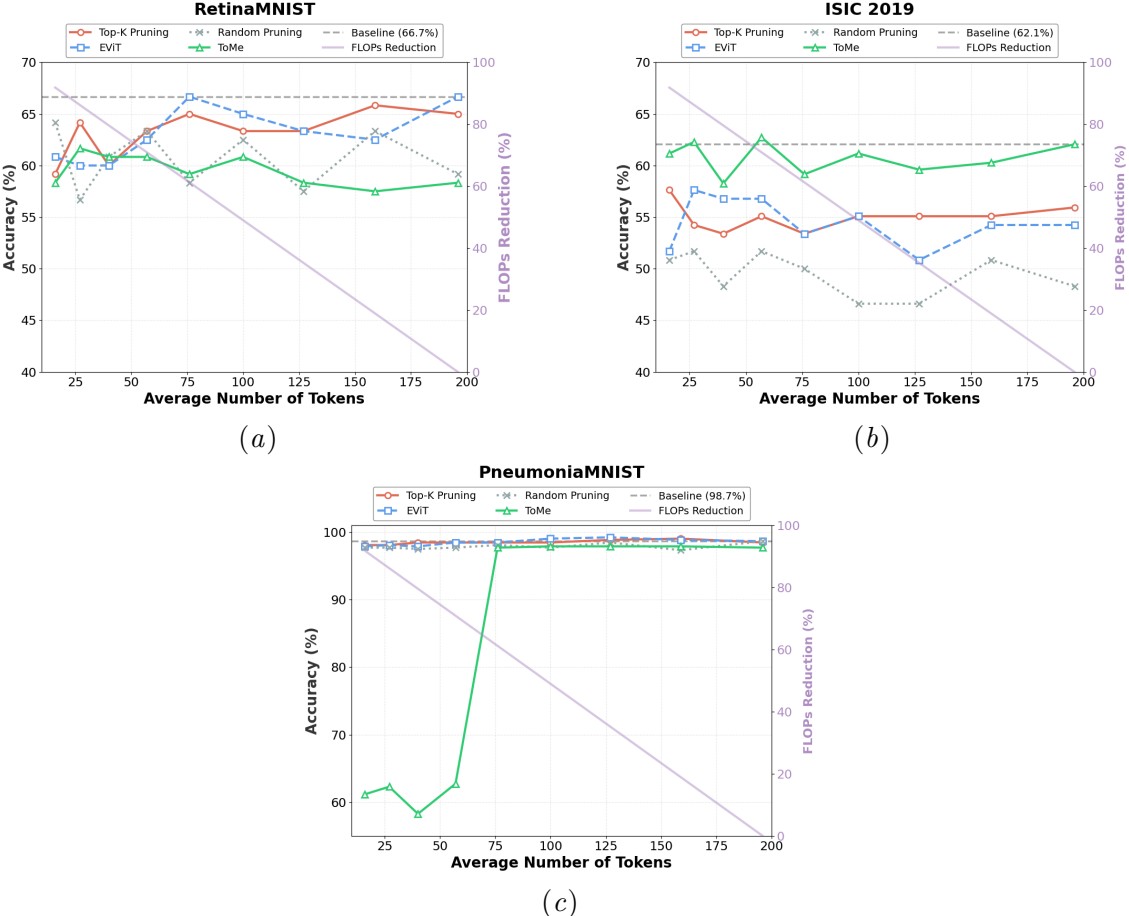

Figure A2: Token reduction strategy comparison across medical imaging datasets. X-axis shows average token count; y-axis shows accuracy (%, left) and FLOPs reduction (%, right, purple line). Methods: Top-K (red circles), EViT (blue squares), Random (gray crosses), ToMe (green triangles). Gray dashed line marks baseline accuracy. (a) RetinaMNIST: EViT maintains consistent performance (60-67% accuracy) across all token budgets. Top-K performs well at higher token counts (65.8% at 159 tokens), while ToMe shows degraded performance comparable to random pruning. (b) ISIC2019: High sensitivity to token reduction across all strategies. ToMe achieves best overall performance (peak 62.7% at 57 tokens). EViT excels at low token budgets while Top-K performs better at high token counts, showing complementary efficiency profiles. (c) PneumoniaMNIST: Highly robust to token reduction across all strategies, maintaining > 97% accuracy down to 16 tokens. EViT, Top-K, and Random show negligible degradation, while ToMe exhibits relative instability below 57 tokens despite maintaining strong absolute performance.

Table A2: **Diagnostic Performance by Strategy and Class.** Comparison of per-class Sensitivity (↑) and Specificity (↑) across five medical imaging datasets. The performance variations highlight dataset-specific biases. **Bolding** indicates the best metric across the four strategies for that specific class/metric. Latency is the achieved runtime (ms).

| Dataset | Class Name | Sensitivity (%) (↑) | | | | | Specificity (%) (↑) | | | | |
|---|---|---|---|---|---|---|---|---|---|---|---|
| | | Baseline | EE-only | TR-only | A-ViT | TR+EE | Baseline | EE-only | TR-only | A-ViT | TR+EE |
| **ISIC 2019** | 0: actinic keratosis | 12.50 | 12.50 | 6.25 | 18.75 | **31.25** | 100.00 | 100.00 | 99.02 | 100.00 | 100.00 |
| | 1: basal cell carcinoma | 87.50 | 87.50 | 87.50 | 87.50 | **93.75** | 94.12 | 95.14 | **97.06** | 95.10 | 87.25 |
| | 2: dermatofibroma | 50.00 | **56.25** | 43.75 | 50.00 | 37.50 | 99.02 | 100.00 | 100.00 | 100.00 | 100.00 |
| | 3: melanoma | 12.50 | 6.25 | 18.75 | 6.25 | **18.75** | 95.14 | 92.16 | 94.12 | **96.08** | **96.08** |
| | 4: nevus | 87.50 | **100.00** | **100.00** | 94.75 | 87.50 | 72.55 | 73.53 | 71.57 | 72.55 | **83.33** |
| | 5: pigmented benign keratosis | 81.25 | **87.50** | **87.50** | 87.50 | 81.25 | 89.22 | **93.14** | 91.18 | 91.18 | 82.35 |
| | 6: seborrheic keratosis | 0.00 | 0.00 | 0.00 | 0.00 | 0.00 | 100.00 | 100.00 | 100.00 | 100.00 | 100.00 |
| | 7: squamous cell carcinoma | 50.00 | 50.00 | **56.25** | **56.25** | 25.00 | 97.06 | 97.06 | 97.06 | **98.04** | **98.04** |
| | 8: vascular lesion | 100.00 | 100.00 | 100.00 | 100.00 | 100.00 | 100.00 | 99.13 | 100.00 | 97.39 | 99.13 |
| **PneumoniaMNIST** | 0: normal | 73.93 | 67.95 | 80.77 | 79.06 | 70.94 | **99.74** | **99.74** | 98.97 | **99.74** | 98.72 |
| | 1: pneumonia | **99.74** | **99.74** | 98.97 | **99.74** | 98.72 | 73.93 | 67.95 | **80.77** | 79.06 | 70.94 |
| **RetinaMNIST** | 0: No DR | **89.66** | 87.93 | 82.76 | 81.03 | 85.06 | 57.96 | 67.70 | 71.24 | 68.14 | **73.45** |
| | 1: Mild DR | 0.00 | 17.39 | **36.96** | 2.17 | 0.00 | 98.59 | 96.61 | 85.88 | **99.72** | 99.15 |
| | 2: Moderate DR | 41.38 | 40.22 | 31.52 | **59.78** | 52.17 | 86.04 | **87.66** | 88.64 | 75.07 | 80.84 |
| | 3: Severe DR | 51.47 | 58.82 | 35.29 | 39.71 | **70.59** | **95.48** | 92.17 | 92.17 | 94.28 | 90.96 |
| | 4: Proliferative DR | **35.00** | 30.00 | 20.00 | 20.00 | 15.00 | 98.42 | 98.16 | 98.42 | 98.42 | **99.74** |
| **PathMNIST** | 0: adipose | **99.33** | 98.58 | 97.38 | 95.44 | 98.43 | 99.50 | 99.67 | 99.52 | 99.66 | **99.91** |
| | 1: background | 100.00 | 100.00 | 100.00 | 100.00 | 100.00 | 99.42 | 99.07 | 99.27 | 99.01 | **100.00** |
| | 2: debris | 95.87 | 97.94 | 99.12 | 95.28 | **97.35** | **99.80** | 99.56 | **99.83** | 98.99 | 99.66 |
| | 3: lymphocytes | 100.00 | 100.00 | 100.00 | 100.00 | 98.90 | 99.88 | 99.48 | 98.75 | 99.25 | **99.89** |
| | 4: mucus | 88.50 | 95.07 | 92.08 | 91.98 | **98.16** | 99.72 | 99.59 | 99.43 | **99.74** | 99.12 |
| | 5: smooth muscle | 93.24 | 87.16 | 85.47 | 85.98 | **92.57** | 97.81 | 98.51 | **98.66** | 97.71 | 98.28 |
| | 6: normal colon mucosa | **97.98** | 95.95 | 90.01 | 84.35 | 93.30 | 98.57 | 98.70 | 98.80 | 99.07 | **99.72** |
| | 7: cancer-associated stroma | 65.80 | 66.51 | 67.70 | 61.06 | **72.92** | **99.87** | 99.73 | 99.39 | 99.75 | **99.87** |
| | 8: colorectal adenocarcinoma epithelium | 96.76 | 94.16 | 95.94 | 95.70 | **98.30** | 99.48 | 99.51 | 99.08 | 97.80 | 99.08 |
| **INSIGHT** | clear | 88.22 | 90.45 | 82.64 | 89.49 | **91.40** | 87.17 | 86.88 | **91.69** | 85.13 | 83.67 |
| | immature cataract | 81.17 | 79.37 | **87.22** | 75.34 | 77.13 | 90.55 | 92.17 | 87.10 | 91.01 | **93.09** |
| | mature cataract | 80.77 | **92.31** | 82.69 | 80.77 | 83.08 | 99.68 | 99.52 | **99.76** | 99.52 | 99.68 |
| | pciol | 92.55 | 94.68 | **97.34** | 93.62 | 93.62 | 99.29 | **99.82** | 99.29 | 98.93 | 99.47 |

Table A3: **Unified Framework Performance Across Datasets Using ViT-S backbone.** Baseline: 196 tokens, 12 layers. EE-only: 196 tokens with dynamic EE (dataset-specific $\theta_{\text{EE}}$). TR-only: EViT with 100 tokens across all layers. TR+EE (Ours, shaded): combines TR and EE. Best results in **bold**.

| Dataset | Strategy | Accuracy (%) | Avg Tokens | Avg Exit Layer | FLOPs (G) |
|---|---|---|---|---|---|
| **ISIC2019** | Baseline | 60.2 | 196 | 12.0 | 4.61 |
| | EE-only | 56.8 (↓3.4pp) | 196 | **3.75** | **1.236** (↓**73.2%**) |
| | TR-only | **61.9** (↑1.7pp) | 100 | 12.0 | 3.003 (↓34.8%) |
| | TR+EE | 56.8 (↓3.4pp) | **68** | 12.0 | 2.117 (↓**54.1%**) |
| **PneumoniaMNIST** | Baseline | 94.2 | 196 | 12.0 | 4.61 |
| | EE-only | 93.9 (↓0.3pp) | 196 | **3.0** | **1.05** (↓**77.2%**) |
| | TR-only | **95.0** (↑0.8pp) | **100** | 12.0 | 3.003 (↓34.8%) |
| | TR+EE | 90.4 (↓3.8pp) | 110 | 5.53 | 1.429 (↓69.0%) |
| **RetinaMNIST** | Baseline | **66.8** | 196 | 12.0 | 4.61 |
| | EE-only | 65.5 (↓1.3pp) | 196 | **6.77** | **1.992** (↓**56.8%**) |
| | TR-only | 63.0 (↓3.8pp) | 100 | 12.0 | 3.003 (↓34.8%) |
| | TR+EE | 62.3 (↓4.5pp) | **68** | 12.0 | 2.117 (↓**54.1%**) |
| **PathMNIST** | Baseline | 93.3 | 196 | 12.0 | 4.61 |
| | EE-only | 93.7 (↓0.5pp) | 196 | **3.13** | **1.082** (↓**76.5%**) |
| | TR-only | **94.6** (↑0.4pp) | **100** | 12.0 | 3.003 (↓34.8%) |
| | TR+EE | 93.5 (↑0.2pp) | 136 | 3.22 | 1.086 (↓76.4%) |
| **INSIGHT** | Baseline | 86.9 | 196 | 12.0 | 4.61 |
| | EE-only | **88.1** (↑1.2pp) | 196 | **3.38** | **1.145** (↓**75.2%**) |
| | TR-only | 87.7 (↑0.8pp) | 100 | 12.0 | 3.003 (↓34.8%) |
| | TR+EE | 86.0 (↓0.9pp) | **70** | 9.46 | 1.979 (↓57.1%) |

