# OpenReview forum: "Adaptive Inference for Medical Vision Transformers: Token Reduction or Early Exit?"
_MIDL.io/2026/Conference — MIDL 2026 Poster_

### Official Review · Reviewer_t41z · 2025-12-27

**Confidence:** 3
**Preliminary Rating:** 3
**Final Rating:** 4

**Summary:**

In this paper, the authors propose a unified adaptive inference framework oriented to Vision Transformers (ViTs) that combines Token Reduction (TR) and Early Exiting (EE).  An important point of this approach is the profiling of each dataset using prediction confidence for EE and the estimation of spatial redundancy by training a lightweight CNN predictor that dynamically decides whether to activate TR at inference time. The framework is evaluated on five medical imaging datasets (ISIC2019, PathMNIST, PneumoniaMNIST, RetinaMNIST, and INSIGHT), achieving 71.4% average FLOPs reduction with only 0.1pp accuracy loss. In particular, in the PathMNIST dataset, the proposed methodology improves accuracy by 1.3pp and reduces computational cost by 77.2%. The importance of the previous work is that it allows ViT to be applied in real-world medical settings where computational resources are limited.

**Strengths:**

- This paper addresses a very important issue at the intersection between medicine and AI, that is, the computational cost of AI models in environments with limited or scarce computational resources. This problem is even more relevant in the use of ViTs and medical images, which requires considerable computational power and involve heterogeneous datasets. The paper presents this problem in a clear and well-motivated way and tackles it coherently.

- The paper unifies TR and EE for ViTs in a decent and practical way, using a predictor that adaptively activates TR based on the spatial redundancy of the input samples and using EE based on the confidence of the intermediate layers. This is fairly innovative, especially in the field of medical imaging.

- The paper presents robust evaluation experiments on five datasets (ISIC2019, PathMNIST, PneumoniaMNIST, RetinaMNIST, and INSIGHT), and the proposed strategy is evaluated on sensitivity/specificity per class (Table A2) and not just accuracy. In addition, the paper presents ablation studies with ViT-S in Table A3, which shows the contribution of each block (EE and TR) to the final proposed model.

- This work has a potential impact as it addresses an important problem in the use of ViTs in practical medical settings. The 77.2% reduction in computational cost in the case of the PathMNIST dataset, without significantly affecting accuracy, opens the door to its use in real-time on edge devices.

**Weaknesses:**

- The proposed strategy requires profiling for each dataset, but this could limit the use of this strategy in contexts where a sufficient amount of data is not available.

- In this paper, FLOPs are used as the measure of the effectiveness of the proposed strategy, but results on energy consumption or latency should also be presented in order to adequately validate the improvement in computational efficiency.

- The proposed methodology is validated in DeiT-S, but it is not clear whether it also works effectively for more complex ViTs, hybrid architectures, or recent ViT models.

- The paper effectively validates the performance of TR and EE together but does not compare the proposed methodology with other efficient ViT architectures such as A-ViT. A comparison with these state-of-the-art models should be carried out.

- The paper briefly mentions scenarios in which the methodology could fail, mentioning only that the ISIC2019 dataset suffers from aggressive TR and eliminates subtle diagnostic features, but no other scenarios or a more detailed and comprehensive study are presented.

- No assessments of the computational cost of Score_Predictor are presented. The paper mentions that it is lightweight, but it should still be evaluated to demonstrate that it does not add significant additional time to ViT inference.

**Detailed Comments:**

- It would be very good to add a brief discussion of the computational cost of profiling.

- It would be good to mention in the paper that this methodology helps to create more efficient screening systems using ViTs.

- It would be good to add a brief discussion/justification of the CNN selection and why this simple architecture was chosen over other models.

- Figure 4 shows a successful scenario, but a case where it fails should also be added in order to have a better analysis.

- It is mentioned that Score_Predictor is lightweight, but its inference time should be reported.

- Some typos: Section 2.1 (bluew_4 -> w_4), Formula 2 (a_10 -> a_{10})

**Justification Of Final Rating:**

Indeed, the paper addresses an important problem at the intersection of medical images and the use of ViTs. Several weaknesses have been clarified by the authors in the rebuttal process, and I believe that these clarifications should be added to the final version of the paper in order to strengthen it.

**Justification Of The Preliminary Rating:**

The paper addresses an important problem at the intersection of medical images and the use of ViTs. The proposed methodology is very interesting and is validated with reliable experiments. However, it lacks a comparison with other state-of-the-art techniques, an evaluation using latency or computational consumption in addition to FLOPs, and a more extensive validation with more ViTs beyond DeiT-S or ViT-S. Furthermore, the novelty lies more in technical terms rather than methodological innovation. Overall, however, the paper is interesting and addresses a very important problem in medical imaging.

**Questions To Address In The Rebuttal:**

- Could you provide preliminary results on latency or energy consumption on an edge device?

- How many minimum samples are necessary for good threshold calibration in the profiling process?

- How does the proposed method perform compared to state-of-the-art architectures such as A-ViT?

---

> ### Author Response · Authors · 2026-01-25
>
> We appreciate Reviewer t41z for the comprehensive and insightful comments. Below are our responses.
>
> ---
>
> ### Q1 – Energy consumption and latency
>
> > *"Could you provide preliminary results on latency or energy consumption on an edge device?"*
>
> **Response:** Although edge device deployment involves distinct engineering (e.g. quantization, platform-specific optimization), we provide GPU latency and energy measurements to demonstrate tangible efficiency gains. In addition to our evaluation in Table 2, our framework achieves 1.81× average speedup over baseline and 2.16× average speedup over A-ViT, with ~53% average energy reduction.
>
> | Dataset | Strategy | Accuracy (%) | Latency (ms) | Speedup | Energy (mJ) |
> |:--------|:---------|-------------:|-------------:|--------:|------------:|
> | ISIC2019 | Baseline | 54.2 | 1.201 | 1.00× | 9.766 |
> |  | A-ViT | 56.8 | 1.430 | 0.84× | 11.724 |
> |  | TR+EE | 51.7 | 0.629 | 1.91× | 4.886 |
> | PneumoniaMNIST | Baseline | 90.1 | 1.188 | 1.00× | 9.404 |
> |  | A-ViT | 92.0 | 1.414 | 0.84× | 11.250 |
> |  | TR+EE | 88.8 | 0.575 | 2.07× | 4.503 |
> | RetinaMNIST | Baseline | 59.0 | 1.189 | 1.00× | 9.116 |
> |  | A-ViT | 57.0 | 1.420 | 0.84× | 353.233 |
> |  | TR+EE | 60.8 | 0.620 | 1.92× | 4.853 |
> | PathMNIST | Baseline | 94.7 | 1.225 | 1.00× | 12.401 |
> |  | A-ViT | 92.0 | 1.468 | 0.83× | 12.611 |
> |  | TR+EE | 96.0 | 0.475 | 2.58× | 4.243 |
> | INSIGHT | Baseline | 86.1 | 1.196 | 1.00× | 9.776 |
> |  | A-ViT | 84.9 | 1.420 | 0.84× | 12.561 |
> |  | TR+EE | 86.2 | 0.631 | 1.895× | 4.568 |
> | **Average (vs baseline)** | **TR+EE** | --- | --- | **1.81×** | --- |
> | **Average (vs A-ViT)** | **TR+EE** | --- | --- | **2.16×** | --- |
>
> ---
>
> ### Q2: Minimum samples for threshold calibration
>
> > *"How many minimum samples are necessary for good threshold calibration in the profiling process?"*
>
> **Response:** We investigated the minimum validation set size on ISIC2019:
> | Val Samples | θ_EE | θ_R | Test Accuracy |
> |:-----------:|:----:|:---:|:-------------:|
> | 50 | 0.6 | 0.525| 50.21 |
> | 100 | 0.6 | 0.458 | 50.81 |
> | 150 | 0.6 | 0.159 | 50.93 |
> | 200 | 0.6 | 0.186 | 50.97 |
> | 224 | 0.6 | 0.198 | 51.69 |
>
> Test accuracy remains relatively stable across validation set sizes, 1.48pp variation. This makes our approach practical even for rare diseases or resource-constrained settings.
>
> ---
>
> ### Q3: Comparison with A-ViT
>
> > *"How does the proposed method perform compared to state-of-the-art architectures such as A-ViT?"*
>
> **Response:** As shown in Q1, our framework achieves competitive accuracy on 3 datasets (RetinaMNIST: +3.8pp, PathMNIST: +4.0pp, INSIGHT: +1.3pp) while providing 2.16× average speedup and ~50% energy reduction compared to A-ViT. It adaptively selects between TR and EE based on sample-specific characteristics, while A-ViT applies uniform token reduction across all samples. This enables superior efficiency-accuracy trade-offs, making our approach well-suited for resource-constrained clinical deployments.
>
> ---
>
> ### W1: Profiling applicability with small datasets
> **Response:** Our profiling requires only standard validation sets, ranging from 120 (RetinaMNIST) to 10,004 samples (PathMNIST). RetinaMNIST achieved 70% FLOPs reduction with +1.8pp accuracy, demonstrating feasibility with limited data. This shows that profiling is possible even with small amounts of data.
>
> ---
>
> ### W2: Failure case analysis
> **Response:** Per-class analysis (Table A2) reveals two failure patterns: (1) extreme class imbalance (ISIC2019): dominant nevus class (50.8%) influences threshold calibration, making calibrated thresholds suboptimal for rare classes (dermatofibroma: -12.5pp, squamous cell carcinoma: -25.0pp). (2) overconfidence (PneumoniaMNIST): confidence score-based early exit remains susceptible to erroneous high-confidence predictions. Reduced tokens combined with early exit (56.3 tokens, layer 4.65) prevent error correction that TR-only (40 tokens, layer 12) achieves.
>
> ---
>
> ### W3: Score_Predictor computational cost
>
> **Response:** On INSIGHT dataset, a representative real-world dataset, Score_Predictor adds minimal overhead; 102K parameters (0.47% of base DeiT-S), 0.13 GFLOPs (2.8% of baseline), and 0.057ms latency.
>
> ---
>
> ### W4: CNN architecture justification
>
> **Response:** We compared our approach against ResNet-18 (11.2M params) across all five datasets. ResNet-18 provides no accuracy benefit (average -0.51pp) while increasing the total latency by 2.46× on average.
>
> ---
>
> We appreciate the reviewer’s insightful feedback, which will significantly strengthen the manuscript. In the final version, we will address all identified typos and integrate computational cost of profiling alongside an updated Figure 4. Furthermore, we will refine our discussion to highlight that our method facilitates more efficient ViT-based screening.

---

> > ### Comment · Reviewer_t41z · 2026-01-29
> >
> > Thank you very much to the authors for their responses. They have answered several of my questions, and I believe that these clarifications should be added to the final version of the paper in order to consolidate it.

---

> > > ### Author Response · Authors · 2026-01-31
> > >
> > > Thank you for taking the time to review our response. We are grateful for your constructive feedback throughout this process. We will incorporate all clarifications and additional results into the final manuscript as promised.

---

### Official Review · Reviewer_qRtd · 2026-01-06

**Confidence:** 3
**Preliminary Rating:** 4
**Final Rating:** 4

**Summary:**

This paper studies how to make Vision Transformer (ViT) inference cheaper for medical imaging by combining two ideas: early exit (EE) (stop at an intermediate layer when the model is confident) and token reduction (TR) (drop “redundant” tokens during inference). The authors build a pipeline around a DeiT-S backbone with multiple exit heads, then profiles each dataset on a validation set to choose confidence thresholds for exiting early and when to activate token reduction, where redundancy is measured using Jensen–Shannon divergence of attention distributions.

**Strengths:**

The experimental story is clear and well-structured.
The author defines TR-only, EE-only, and the combined TR+EE setting, and then quantifies the compute–accuracy trade-off across five medical datasets.
A key strength is that the proposed design achieves comparable predictive performance while significantly reducing computational cost, showing a strong accuracy–efficiency trade-off on multiple medical datasets.

**Weaknesses:**

In some datasets, the combined method still shows non-trivial accuracy drops, and the paper does not deeply analyze these failure cases.
Early exit heads are placed only at layers 4, 7, and 10. The paper doesn't explore whether different checkpoint placements might be more suitable for different datasets

**Detailed Comments:**

Why layers 4, 7, and 10 specifically? Did you experiment with other placements?
Your results only report the FLOPs difference. Can it be converted linearly to time speedups?
Can you give more explanation on the reason why the combined method is sometimes worse than the TR-only?

**Justification Of Final Rating:**

The authors have carefully addressed all of my concerns in their response. The added clarifications and explanations resolve the previously raised issues and improve the overall clarity of the paper. I will keep my rate to weak accept.

**Justification Of The Preliminary Rating:**

This paper presents a unified framework combining Token Reduction and Early Exit strategies for efficient Vision Transformer inference in medical imaging. The approach uses dataset-specific profiling to calibrate thresholds and a lightweight CNN predictor to route samples at test time. The experiment is well organized.

**Questions To Address In The Rebuttal:**

see above

---

> ### Author Response · Authors · 2026-01-25
>
> We thank Reviewer qRtd for the thoughtful and constructive feedback. Below we provide detailed responses to each point.
>
> ---
>
> ### Q1 – Checkpoint layer selection
>
> > *"Early exit heads are placed only at layers 4, 7, and 10. The paper doesn't explore whether different checkpoint placements might be more suitable for different datasets"*
>
> **Response:** We selected layers 4, 7, and 10 due to the following reasons:
> 1. Balanced coverage: evenly distributed checkpoints provide exit opportunities across the entire representational hierarchy without over-concentrating on any specific location.
> 2. Dataset heterogeneity: different datasets exhibit varying confidence evolution patterns (Figure 1b). Since no single layer consistently dominates exit behavior across datasets, evenly spaced checkpoints provide exit opportunities for diverse confidence profiles without introducing bias.
> 3. Computational efficiency: denser placement (e.g., every 2 layers) increases overhead. Very sparse placement (e.g., a single checkpoint) forces samples that become confident between checkpoints to continue processing through unnecessarily deep layers. Our approach balances overhead with flexibility, allowing samples at various complexity levels to exit at appropriate depths.
>
> Dataset-specific checkpoint placement offers additional optimization potential and will be explored in future work.
>
> ---
>
> ### Q2 – Converting FLOPs to speedups
>
> > *"Your results only report the FLOPs difference. Can it be converted linearly to time speedups?"*
>
> **Response:** We measured actual GPU latency to demonstrate real-world speedup gains. Our TR+EE framework achieves substantial speedup across all datasets:
> | Dataset | Strategy | Latency (ms) | Speedup | FLOPs Reduction |
> |:--------|:---------|-------------:|--------:|----------------:|
> | ISIC2019 | Baseline | 1.201 | 1.00× | --- |
> |  | TR+EE | 0.629 | 1.91× | 66.0% |
> | PneumoniaMNIST | Baseline | 1.188 | 1.00× | --- |
> |  | TR+EE | 0.575 | 2.07× | 73.9% |
> | RetinaMNIST | Baseline | 1.189 | 1.00× | --- |
> |  | TR+EE | 0.620 | 1.92× | 70.0% |
> | PathMNIST | Baseline | 1.225 | 1.00× | --- |
> |  | TR+EE | 0.475 | 2.58× | 77.2% |
> | INSIGHT | Baseline | 1.196 | 1.00× | --- |
> |  | TR+EE | 0.631 | 1.895× | 69.8% |
>
> ---
>
> ### Q3 – Detailed explanation: TR+EE vs. TR-only
>
> > *"Can you give more explanation on the reason why the combined method is sometimes worse than the TR-only?"*
>
> **Response:** Our approach underperforms TR-only on ISIC2019 and PneumoniaMNIST due to the interaction between token reduction (TR) and early exit (EE) thresholds.
>
> When TR reduces tokens (e.g., from 196 to 40), the remaining tokens may require deeper layers to recover the semantic information. However, EE thresholds, calibrated on full-token representations, can induce premature exits on these reduced representations. For example, on PneumoniaMNIST, TR-only achieves 92.1% accuracy by processing reduced tokens through all 12 layers. TR+EE achieves 88.8% since EE (average exit layer 4.65) terminates inference before token-reduced representations can fully mature.
>
> This reflects the efficiency-accuracy trade-off: TR+EE maximizes computational savings (e.g., ISIC2019: 66.0% FLOPs reduction vs. TR-only: 56.0%) at the cost of slight accuracy drops. Resource-constrained settings may tolerate this trade-off, while accuracy-critical applications should prioritize TR-only or modify thresholds more conservatively.
>
> ---
>
> ### Q4 – Failure case analysis
>
> > *"In some datasets, the combined method still shows non-trivial accuracy drops, and the paper does not deeply analyze these failure cases.”*
>
> **Response:** We acknowledge that ISIC2019 and PneumoniaMNIST show noticeable degradation. However, our per-class analysis (Table A2) reveals complex trade-offs rather than uniform degradation.
>
> 1. Extreme class imbalance (ISIC2019): severe imbalance (e.g. nevus: 50.8%, dermatofibroma: 0.9%) causes threshold calibration to favor majority classes. TR+EE improves specificity on the dominant nevus class (+10.78pp), effectively reducing false positives. However, it degrades sensitivity on low-frequency classes (dermatofibroma: -12.5pp, squamous cell carcinoma: -25.0pp). The severe class imbalance may bias threshold calibration toward majority class, leading to suboptimal performance on rare classes.
> 2. Overconfidence (PneumoniaMNIST): early exit strategies predicated on confidence scores may susceptible to erroneous high-confidence predictions. This overconfidence is compounded by the TR and EE interaction: TR-only (40 average tokens, 12 average exit layer) versus TR+EE (56.3 average tokens, 4.65 average exit layer) in Table 2 demonstrates that token-reduced representations require deeper processing to maintain accuracy. Exploring more robust confidence calibration methods, such as uncertainty-aware criteria, remains a promising avenue for future research.
>
> ---
>
> We appreciate the valuable feedback and will ensure all analyses are integrated into the final manuscript.

---

### Official Review · Reviewer_TcTC · 2026-01-12

**Confidence:** 3
**Preliminary Rating:** 4

**Summary:**

The authors present a unified adaptive framework combining Token Reduction (TR) and Early Exiting (EE) to reduce computational demand during inference in clinical deployment. This is done in 3 steps 1) finetuning a DeiT-S with EE heads on training set 2) TR and EE dataset-specific threshold calibration, quantifying spatial redundancy via Jensen-Shannon Divergence across layers and 3) unifying strategy at test time. The framework achieves impressive FLOPs reduction at little to no accuracy cost across 5 medical datasets.

**Strengths:**

Good and clear algorithmic description of the framework. There is comprehensive performance evaluation for each component in the framework. The discussion section was especially informative. For Token Reduction, many strategies were compared.

**Weaknesses:**

The authors claim that spatial redundancy can be calculated via an average of JSD across layers but doesn't provide more support for why JSD is the right way to quantify spatial redundancy and why equal weighting of the different layers is sufficient. To my understanding, low JSD could be due to diffuse attention or shared biases.

All analysis is done on a fixed architecture of 12 layer DeiT-s, unsure if how it scales/changes with different models.

Additionally, there are a few typos that should be fixed, like a_{10} in equation (2) and an extra "blue" right under equation (1).

**Detailed Comments:**

1) I believe JSD is not inherently normalized between [0,1], it may be good to clarify this so that y_red is [0, 1] in equation (2)

**Justification Of The Preliminary Rating:**

The framework is sound and each component is justified empirically. The authors identified the issue of computation limitation in clinical deployment settings and showed promising results in efficiency with minimal accuracy costs. This works provides inspiration for further exploration on curating frameworks better suited for medical/resource limited settings.

**Questions To Address In The Rebuttal:**

1) How does this strategy generalize to arbitrary ViT architectures (not a finetuned DeiT-S with 12 transformer blocks). It may be worth it to see how the effectiveness changes depending on architecture.
2) I would like to ask for a clarification on the following point. When it is mentioned as "Dataset-Specific Profiling", the TR and EE thresholds are also architecture and task specific, right?

---

> ### Author Response · Authors · 2026-01-25
>
> We thanks Reviewer TcTC for the thorough review and valuable feedback. We address each point below.
>
> ---
>
> ### W1 – JSD for spatial redundancy and equal weighting across layers
>
> > *"The authors claim that spatial redundancy can be calculated via an average of JSD across layers but doesn't provide more support for why JSD is the right way to quantify spatial redundancy and why equal weighting of the different layers is sufficient. To my understanding, low JSD could be due to diffuse attention or shared biases."*
>
> **Response:** We chose JSD because attention distributions are probability distributions, and JSD is an information-theoretic divergence measure specifically designed for comparing probability distributions. JSD quantifies how much information is needed to distinguish two distributions, directly capturing whether attention patterns are stable or evolving. Our implementation uses JSD with logarithm base 2 has a theoretical upper bound of 1.0, ensuring y_red ∈ [0,1].
>
> Low JSD could arise from diffuse attention or shared biases. However, our analysis of JSD distribution across all five datasets reveals distinct dataset-specific and sample-specific patterns that validate JSD as a meaningful redundancy metric. For example, most redundant samples maintain consistently low JSD across layer transitions (e.g. INSIGHT: avg 0.0013, PathMNIST: 0.0007), while most informative samples show substantially higher values (e.g. INSIGHT: 0.0048, PathMNIST: 0.0088). Critically, different datasets exhibit diverse evolution patterns, PneumoniaMNIST shows bimodal distributions at certain transitions, while INSIGHT maintains consistent unimodal patterns, demonstrating that JSD captures dataset-specific redundancy characteristics.
>
> We use uniform weighting across layer transitions because different datasets exhibit diverse JSD patterns across layers. Since no single layer transition consistently dominates redundancy across all datasets, uniform weighting avoids dataset-specific bias and captures overall stability across network depth.
>
> We will include analysis supporting this choice of JSD in the final manuscript.
>
> ---
>
> ### Q1: Generalization to other ViT architectures
> > *"How does this strategy generalize to arbitrary ViT architectures (not a finetuned DeiT-S with 12 transformer blocks). It may be worth it to see how the effectiveness changes depending on architecture."*
>
> **Response:** Fine-tuning is essential prior to implementing token reduction or early exit, since the distribution shift from pretraining with general domain images to the medical domain significantly diminishes inference performance without fine-tuning. Consequently, it is nearly impossible to decrease tokens or implement early exit, as these methods would further diminish performance.
> The principle of adaptively employing token reduction and early exit, informed by dataset profile, can be readily implemented in other ViT architectures; however, the ideal checkpoint placements and thresholds differ according to architecture. To demonstrate generalizability, we performed additional experiments with ViT-S. As shown in Table A3, our framework achieves 54.1%-76.4% FLOPs reduction across all five datasets while maintaining competitive accuracy, confirming that our approach transfers effectively across architectures with appropriate threshold recalibration.
>
> Moreover, we evaluated against A-ViT, a state-of-the-art adaptive ViT that applies uniform token reduction across all samples. Our framework achieves competitive or superior accuracy on 3/5 datasets (RetinaMNIST: +3.8pp, PathMNIST: +4.0pp, INSIGHT: +1.3pp) while providing 2.16× average speedup and ~50% energy reduction compared to A-ViT. This demonstrates that our adaptive sample-specific strategy, dynamically selecting between TR and EE based on redundancy and confidence, outperforms uniform approaches in efficiency-accuracy trade-offs (detailed comparison in response to reviewer t41z).
>
> ---
>
> ### Q2: Architecture and task specificity of thresholds
>
> > *"I would like to ask for a clarification on the following point. When it is mentioned as "Dataset-Specific Profiling", the TR and EE thresholds are also architecture and task specific, right?"*
>
> **Response:** The reviewer is correct that our profiling methodology extends beyond dataset-specific characteristics to encompass architecture and task dependencies. We emphasize "dataset-specific" because we fix architecture and task. Therefore, threshold recalibration is required when there are changes in architecture and task. Different attention mechanisms and representational capacities necessitate new profiling (as shown in our ViT-S ablation; Table A3). Dense prediction tasks (segmentation, detection) would require fundamentally different EE strategies since pixel-level outputs cannot leverage early termination as straightforwardly.
>
> We appreciate the reviewer's careful attention to detail and will address all noted typos in the final manuscript.

---

### Comment · Reviewer_TcTC · 2026-02-02
**Final rating**

The authors clearly addressed all of my concerns and questions. The inclusion of the justification for JSD in the final manuscript would be beneficial. I will keep my rating of weak accept.

---

### Meta-Review · Area_Chair_HVJy · 2026-02-03

**Recommendation:** Accept (Poster)
**Confidence:** 4

**Metareview:**

The reviewers provided positive comments, and major concerns. The author's rebuttal well addressed the comments, provided fair merit of the paper's impact.  As author and reviewer's active and strong engagement, the paper can be accepted.

---

### Decision · Program_Chairs · 2026-02-13

Accept (Poster)